# Association of Healthy Predominantly Plant-Based Diet with Reduced Cardiovascular Disease Incidence and Mortality and Development of Novel Heart-Protective Diet Index [note 1]

**DOI:** 10.3390/nu17162675

**Published:** 2025-08-19

**Authors:** Tian Wang, Andrea Nova, Sophie Cassidy, Katherine M. Livingstone, Teresa Fazia, Sayan Mitra, Cynthia M. Kroeger, Andrius Masedunskas, Luisa Bernardinelli, Walter C. Willett, Luigi Fontana

**Affiliations:** 1Charles Perkins Centre, The University of Sydney, Sydney, NSW 2050, Australia; sophie.cassidy@sydney.edu.au (S.C.); sayan.mitra@sydney.edu.au (S.M.); cynthia.kroeger@sydney.edu.au (C.M.K.); andrius.masedunskas@sydney.edu.au (A.M.); luigi.fontana@sydney.edu.au (L.F.); 2Faculty of Medicine and Health, The University of Sydney, Camperdown, NSW 2050, Australia; 3Department of Brain and Behavioral Sciences, University of Pavia, 27100 Pavia, Italy; teresa.fazia01@ateneopv.it (T.F.); luisa.bernardinelli@unipv.it (L.B.); 4Institute for Physical Activity and Nutrition, School of Exercise and Nutrition Sciences, Deakin University, Burwood, VIC 3125, Australia; k.livingstone@deakin.edu.au; 5Harvard T. H. Chan School of Public Health, Boston, MA 02115, USA; 6Department of Endocrinology, Royal Prince Alfred Hospital, Sydney, NSW 2050, Australia

**Keywords:** cardiovascular diseases, diet quality, incidence, mortality, plant-based diet

## Abstract

**Background:** Previous research examining the effects of omnivorous and plant-based diets on cardiovascular disease (CVD) outcomes has produced inconsistent findings, and the importance of diet quality is overlooked. Our study aimed to develop a novel heart-protective diet index to assess the association of a high-quality, predominantly plant-based diet—including fish, eggs, and low-fat dairy products—with CVD incidence and mortality. **Methods:** This study included 192,274 participants in the UK Biobank (mean age: 56.3 ± 7.9 years) without CVD at baseline who completed a 24 h recall Oxford WebQ questionnaire. Using the Oxford WebQ questionnaire, we developed and validated a novel heart-protective diet score (HPDS) based on 22 food groups. Cox proportional hazard models were used to study the associations between HPDS and outcomes. **Results:** During a median follow-up of 12.3 years, 20,692 CVD events and 1131 CVD deaths were observed. After adjusting for demographics, Townsend deprivation index, lifestyle, and history of chronic diseases, participants in the top HPDS quartile were at lower risk, compared to those in the bottom quartile, of overall CVD (HR: 0.92 [95%CI: 0.88, 0.95]), IHD (HR: 0.89 [95%CI: 0.84, 0.94]), MI (HR: 0.85 [95%CI: 0.77, 0.94]), and HF (HR: 0.86 [95%CI: 0.77, 0.95]). **Conclusions:** Adherence to a healthy, predominantly plant-based heart-protective diet rich in non-starchy vegetables, fruits, wholegrains, fish, eggs, and low-fat dairy products is associated with a reduced cardiovascular disease incidence and mortality. Further research in diverse ethnic populations is necessary to examine the reproducibility of our findings and enhance generalizability.

## 1. Introduction

Despite significant progress in the management of cardiovascular diseases (CVDs), they remain the leading cause of disease burden and mortality globally [1]. A key contributing factor to the heightened risk of CVDs is a poor-quality diet. In contrast, adherence to heart-healthy dietary patterns has consistently demonstrated a positive impact on cardiovascular health [2,3]. These patterns emphasize the consumption of a wide variety of vegetables, fruits, minimally processed whole grains, healthy protein sources (primarily plant-based foods, fish, eggs, and low-fat dairy products), and liquid plant oils. Additionally, they discourage the intake of red meat, highly processed foods that are rich in empty calories and unhealthy fats, added salt and sugars, and alcohol [2,3,4].

Previous research examining the effects of omnivorous and plant-based diets on CVD outcomes has produced inconsistent findings. Some studies have suggested that incorporating fish into a plant-based diet, as seen in the pesco-vegetarian diet, may provide protection against CVD outcomes [5,6]. However, conflicting results have indicated no significant difference or even an elevated risk of CVD among pesco-vegetarians and lacto-ovo-vegetarians [7,8].

Several studies failed to take into account the importance of diet quality, which is a critical factor influencing cardiometabolic health. Individuals adhering to “unhealthy” plant-based diets that prioritize refined grains, high-sodium preserved vegetables, deep-fried foods, and sweetened beverages have an elevated CVD risk and mortality [9,10]. Moreover, certain prospective studies relied on dietary questionnaires with only four questions to determine participants’ vegetarian status and classify them into various diet groups. To address this research gap, a thorough evaluation of diet quality is warranted.

Diet quality indices are commonly used by researchers in observational studies to assess the quality of plant-based diets [10,11]. Marchese et al. [12] recently reviewed 35 plant-based diet quality indices, of which 19, 9, and 5 indices were developed based on food frequency questionnaire (FFQ), semi-FFQ, and 24 h recall, respectively. To allow for a thorough evaluation of diet quality, detailed dietary data is needed to reflect individuals’ adherence to a plant-based diet. Past research has mixed definitions of plant-based diets, as reflected in the existing plant-based diet quality indices. For example, hPDI and uPDI [9,10] only evaluate the quality of plant-based food items but assign all animal food items negative scores, while the other three indices (Comprehensive Diet Quality Index (cDQI) [13], Diet Quality Index Associated to the Digital Food Guide (DQI-DFG) [14], and Global Diet Quality Score (GDQS) [15]) consider the healthiness of both animal-based and plant-based foods. Among them, discrepancies in the categorization of food groups are identified in their scores for red meats, eggs, processed meats, full-fat dairy products, tubers and roots, refined carbohydrates, and sugars and sweets. Yet, none of these scores fully reflect the diet recommended by the American Heart Association (AHA), the Australian Heart Foundation (AHF), and the European Society of Cardiology (ESC)/European Atherosclerosis Society (EAS) guidelines [16,17,18] for the prevention and management of CVD.

To overcome some of these limitations, we aim to (1) develop and validate a novel healthy predominantly plant-based heart-protective diet score (HPDS); (2) evaluate the overall quality of the diet and explore whether adherence to this diet could potentially lower the risk of cardiovascular events among individuals in the UK Biobank (UKBB) population.

## 2. Methods

### 2.1. Study Population

The UK Biobank is an extensive prospective cohort study conducted within the general population, which enrolled over 500,000 individuals aged 40–69 years from 2006 to 2010 [19].

A detailed description of dietary data collection and the Oxford WebQ [20] can be found in Appendix A. In the present study, we used data collected by the Oxford WebQ (24 h dietary measurement). We included participants (n = 192,274) who (i) had completed at least one Oxford WebQ between 2009 and 2012; (ii) had relevant clinical records and death registry data available to measure our outcomes of interest (described below). Individuals excluded from our study are described in the Appendix A.

We calculated the “heart-protective diet score at baseline” and the “averaged heart-protective diet score” based on repeated measurements and found a moderate correlation (Pearson correlation coefficient = 0.53; *p*-value < 0.001) between them. Sensitivity analyses were further conducted using 24 h recall data from the first three occasions, and results were almost identical to the findings generated from the single 24 h recall (Appendix A). Therefore, for participants who completed the dietary measurement multiple times, we used the earliest data to maximize the follow-up duration.

### 2.2. Measurement of Dietary Intake and Development of the Plant-Based Heart-Protective Diet Scores

Diet quality is essential in cardiovascular prevention and management, which has been emphasized by multiple international CVD clinical guidelines, including ESC/EAS [16,17,18], American College of Cardiology (ACC)/AHA, and AHF [21]. These guidelines consistently recommend a predominantly healthy plant-based diet, rich in a diverse range of vegetables, fruits, whole grains, nuts, healthy protein sources (such as legumes and fish), while they discourage the consumption of red meats, sweets, added sugars, and added salt. To assess the effect of adherence to such a diet on cardiovascular outcomes, a thorough evaluation of diet quality is warranted.

Diet quality indices are commonly used by researchers to assess the quality of plant-based diets [10,11]. Both FFQ and 24 h recall are commonly used dietary assessment approaches. However, as highlighted by Nutritools and Dietary Assessment Guidelines [22], short FFQs are not reliable for evaluating the total diet. To allow a thorough evaluation, detailed dietary data is needed to reflect individuals’ adherence to a plant-based diet. Appendix A summarize the current main plant-based diet quality indices utilizing data from 24 h recall, and explain the rationale of developing a novel heart-protective diet score.

To address the current research gaps in plant-based diets and cardiovascular events, a new score designed specifically for cardiovascular health, incorporating current guidelines recommendations, is warranted. We therefore developed a “heart-protective diet score” (HPDS) based on three previously validated plant-based diet indices (PDI): PDI, hPDI, and uPDI [9,10], which distinguishes the healthiness of plant-based foods based on the existing evidence of those foods with cardiometabolic diseases. Our HPDS additionally assesses the healthiness of animal-based foods following the most updated dietary recommendations in the management of cardiovascular health with a strong evidence base. These include the 2021 ACC/AHA, AHF, ESC/EAS guidelines and the Global Burden of Disease Nutrition and Chronic Disease Expert Group [21]. This scoring system was developed based on existing evidence [23,24] regarding the associations between specific foods and nutrients and their impact on cardiometabolic outcomes. The HPDS emphasizes the consumption of a diverse range of vegetables, fruits, whole grains, nuts, healthy protein sources (such as legumes and fish), while it discourages the consumption of red meats, sweets, added sugars, and added salt. Highly processed vegetarian alternatives [25,26,27], vegetable fats [28,29], salted nuts [30] and alcoholic beverages [31,32] are not included in the HPDS given the large variety in healthiness of novel food products (e.g., sodium contents vary largely in different vegetarian products) and mixed evidence of their associations related to different health outcomes. To derive the HPDS, we categorized the food and drink items into 22 food groups (Table 1) based on the current dietary recommendations by the 2021 ACC/AHA, AHF, ESC/EAS guidelines and the Global Burden of Disease Nutrition and Chronic Disease Expert Group and the dietary data collected by the Oxford WebQ, and ranked consumption into quintiles. A detailed description of the calculation of HPDS, as well as the validity and reliability (including construct validity, criterion validity, reliability, and content validity), was evaluated and described in the Appendix A.

### 2.3. Cardiovascular Disease Incidence and Death

The primary outcomes included the CVD incidence and mortality. We defined CVD outcomes based on the earliest diagnosis recorded according to the International Classification of Disease 10th revision (ICD10) (see Appendix A) for the following diseases: (i) ischemic heart disease (IHD); (ii) myocardial infarction (MI); (iii) stroke; (iv) heart failure; (v) arrhythmia and conduction disorders including atrial fibrillation. To ascertain these diagnoses, we obtained data from linked sources including hospital admissions, primary care records, death registries, and self-reported codes (Appendix A) [33,34,35,36]. Furthermore, we incorporated diagnoses for MI, stroke, and IHD obtained from the Electronic Health Records (EHRs) maintained by General Practitioners in primary care [37]. Participants were followed up from the date of their initial assessment (March 2006 to October 2010), until the date of outcome diagnosis, date of death, date of loss to follow-up, or the end of follow-up (31 December 2021), whichever came first.

### 2.4. Cardiometabolic Outcomes

Secondary outcomes included incidence of CVD risk factors: (i) overweight/obesity; (ii) hypertension; (iii) hyperlipidaemia; (iv) hyperglycaemia or type II diabetes mellitus (T2DM). Similar to the primary outcomes, the diagnoses for these conditions were obtained using the same methods as described earlier, and self-reported disease status supplemented hypertension and T2DM (Appendix A) [35,38,39,40,41]. Moreover, diagnoses for hypertension and T2DM were also extracted from the EHRs [37]. We excluded participants who had prevalent cases of overweight and obesity (n = 3799), hypertension (n = 42,788), hyperglycemia or T2DM (n = 3535), and hyperlipidemia (n = 189) at the time of recruitment. We then calculated person-years as previously described for the primary outcomes.

### 2.5. Study Design

This is a prospective cohort study, where participants were followed up from the date of their initial assessment (March 2006 to October 2010), until the date of outcome diagnosis, date of death, date of loss to follow-up, or the end of follow-up (31 December 2021), whichever came first.

### 2.6. Statistical Analysis

Characteristics of the cohort within quartiles of HPDS were reported for continuous (mean (standard deviation [SD])) and categorical variables (%). The associations between HPDS and the outcomes were evaluated using a time-to-event analysis. Cox proportional hazard regression models, with age at recruitment as the underlying timescale, were implemented to estimate hazard ratios (HRs) and 95% confidence intervals (CIs) for the associations between HPDS in quartiles with primary and secondary outcomes. For these analyses, we used the *survival* R package. Linear trend was tested keeping HPDS in the model as a continuous variable. To account for confounding, the model was adjusted for variables associated with both diet consumption and CVD risk, including sex, ethnicity, sociodemographic factors, lifestyle (e.g., physical activity, smoking habits, alcohol intake), history of hyperglycaemia/T2DM diagnosis, and history of cancer diagnosis. A thorough description of covariates can be found in the Appendix A. Potential multicollinearity of covariates was tested, and the highest correlation was 0.33 between education and income. Therefore, multicollinearity was not a concern. The proportional hazard assumption was checked by tests based on Schoenfeld residuals (*cox.zph()* function), and stratification factors (*strata()* function) were applied to confounders not respecting the assumption, i.e., *p* < 0.05. To provide information on the impact of each food group on CVD incidence and mortality, we further implemented the same Cox model, considering each food group as a variable in the model. Sensitivity analyses were performed, excluding self-reported diagnoses. As secondary analyses, we examined whether sex and Townsend deprivation index (TDI) modified the association between HPDS and primary/secondary outcomes using multiplicative interaction terms. For significant interactions, subgroup analyses were conducted stratifying by sex and TDI median value. To account for multiple testing, we applied a False Discovery Rate (FDR) of 0.05 with Benjamini–Hochberg adjustment. Missing data for confounding variables were imputed using multiple imputations with the predictive mean matching method on ten datasets using the *mice* R package. Rubin’s rule was used to aggregate the results and obtain HRs with 95%CIs. RStudio 2022.02.2+485 was used for all analyses.

## 3. Results

### 3.1. Sociodemographic and Dietary Characteristics of Participants

A total of 192,274 participants (mean age: 56.3 ± 7.9 years, 43.3% males) were included (Table 2; Appendix A). During a median follow-up period of 12.3 years (interquartile range: 11.7, 13.2), we documented 20,692 (10.8%) incident cases of CVD (Figure 1: 11,241 IHD, 3452 MI, 3103 stroke, 3189 HF, and 8913 AF) with an average age at diagnosis of 67.5 ± 7.4 years. Moreover, 1131 participants (0.59%) died from CVD (Figure 2: 513 IHD, 292 MI, 351 stroke, 141 HF). Table 2 shows participants’ baseline characteristics for quartiles of HPDS. The calculated HPDS had a normal distribution (Appendix A), with a mean value of 2.78 ± 7.60 (ranging from −30.00 to 36.92). Individuals with higher HPDS were older, more likely to be female, non-current smokers, physically active, had a higher likelihood of having a university degree, higher income, and were more likely to take dietary supplements, demonstrating the construct validity of HPDS in differentiating populations by sociodemographic characteristics.

Table 3 shows food groups and nutrient intake by quartiles of HPDS. Compared to Q1, individuals with the highest HPDS consumed more non-starchy vegetables, fruits, wholegrains, legumes and beans, uncoated fish and seafood, and tea and other low-calorie drinks, resulting in a lower intake of energy (−200 kcal/d), total fat (−10 g/d), saturated fat (−8 g/d) and sodium (−310 mg/d, 15% lower), and a higher intake of dietary fiber (+8 g/d), vitamins (vitamin C, folate, beta carotene, vitamin E, and vitamin B12) and minerals (potassium, magnesium, and iron). These results illustrate the construct validity of HPDS, in which HPDS is positively correlated with the intake of key health-promoting nutrients and food groups. In contrast, individuals with the lowest HPDS consumed more refined grains and cereals, meat, poultry, processed meat, sweets, desserts, cookies, and pastries.

### 3.2. Higher Healthy Plant-Based Diet Score Is Associated with Lower Cardiovascular Diseases Incidence and Mortality

The pooled results from the Cox proportional hazard models are presented in Figure 1 and Figure 2, and Appendix A. Individuals in the highest HPDS quartile (Q4) experienced a statistically significant 8% decreased risk in overall CVD incidence (Figure 1). Specifically, compared to Q1, the highest HPDS was significantly associated with a 15% decreased risk of MI incidence, a 14% decreased risk of HF incidence, and an 11% decreased risk of IHD incidence. These findings demonstrated the criterion validity of our HPDS.

A stronger protective and statistically significant association was observed for individuals in the highest HPDS quartile (Q4) compared to Q1 for CVD mortality outcomes. These included a 23% decreased risk in overall CVD mortality, a 48% decreased risk in HF mortality, a 34% decreased risk in MI mortality, and a 12% decreased risk in IHD mortality.

### 3.3. Higher Healthy Plant-Based Diet Score Is Associated with Lower Incidence of Cardiometabolic Abnormalities

Comparing individuals in the highest HPDS quartile (Q4) to Q1, we observed statistically significant protective associations with cardiometabolic health (Figure 3, Appendix A). Specifically, individuals in Q4 had a 23% lower risk of being overweight and obese, a 19% lower risk of developing hyperlipidemia, a 17% lower risk of developing hyperglycemia and T2DM, and a 7% lower risk of developing hypertension compared to individuals in Q1. These findings suggest that individuals in the highest HPDS quartile have a significantly reduced risk of experiencing cardiometabolic abnormalities compared to those in the lowest quartile.

### 3.4. Moderation Effects of Sex and Townsend Deprivation Index, and Sensitivity Analyses

In our analysis, we identified statistically significant multiplicative interactions (*p* < 0.05 after multiple testing correction) between HPDS and sex, as well as between HPDS and the TDI, for various cardiometabolic outcomes and atrial fibrillation incidence. When stratifying the data by sex (Appendix A), we observed a stronger protective association for females with higher HPDS values compared to males across several outcomes, namely: CVD, IHD, HF, and AF incidence, overweight and obesity, hypertension, hyperglycemia, and T2DM incidence. These results suggest that a healthy plant-based diet may have a more pronounced beneficial impact on these outcomes for females. When stratifying the analysis by the median value of the Townsend deprivation index (2.6) (Appendix A), we did not find any statistically significant difference between the two groups. Furthermore, sensitivity analyses comparing outcomes with and without self-reported diagnoses (Appendix A) yielded consistent findings, indicating the robustness of our results. Finally, in Appendix A, we reported the associations between each of the 22 food groups with CVD incidence and mortality.

## 4. Discussion

Previous validated plant-based diet indices only evaluate the healthiness of plant-based foods, but assign negative scores to animal-based foods. To the best of our knowledge, only three indices include the evaluation of animal-based foods, but their categorization did not fully reflect the clinical guidelines and recommendations by the AHA, HF, and ESC/EAS. To address this and evaluate the effect of following the recommended diet on the risk of developing cardiovascular outcomes, we have developed and validated a novel plant-based diet index, which utilizes detailed dietary data and assigns positive scores to fish and seafood, eggs, and reduced fat dairy products. In this large cohort study of middle-aged and older adults, a higher adherence to a healthy plant-based heart-protective diet rich in non-starchy vegetables, fruits, whole grains, legumes, and fish was associated with a reduced risk of CVD and lower mortality rates. Those who closely followed this dietary pattern had a 49% lower risk of mortality from heart failure, along with a 32–35% lower risk of mortality from IHD and MI. Additionally, the healthy plant-based diet score was linked to a lower incidence of overweight, hyperlipidemia, hypertension, and hyperglycemia. These findings underscore the potential benefits of adopting a healthy plant-based diet in reducing the risk of CVDs, mortality, and improving overall cardiometabolic health.

Our study demonstrates that a higher adherence to our healthy plant-based diet score leads to a greater reduction in cardiovascular morbidity and mortality compared to previous studies focusing solely on vegetarian and vegan diets [9,42,43,44]. Past research has demonstrated the protective effects of plant-based diets on cardiometabolic risk factors [45,46]. A recent meta-analysis showed that following a vegetarian diet for an average of six months is effective in improving low-density-lipoprotein cholesterol LDL-C, glycated haemoglobin HbA1c, and body weight in individuals at high risk of CVDs [45]. While vegetarian and vegan diets have shown positive associations with cardiovascular health, the comprehensive nature of our HPDS, which includes a diverse range of plant foods with fish and seafood while limiting processed foods, refined carbohydrates, and unhealthy fats, appears to have even more pronounced effects on cardiovascular and metabolic health. By considering the individual components of the diet and integrating them into a simplified scoring system, the combined impact becomes more evident.

Compared to the Mediterranean diet, which typically includes both refined and whole grains, moderate alcohol intake, and discourages dairy products and potatoes [47], our HPDS highlights the significant impact of a healthy plant-based diet that prioritizes minimally processed foods. Notably, a recent UK Biobank study using the Mediterranean diet score reported only a small 7% reduction in the risk of stroke and no significant effect on MI [48]. In that study, both refined and whole grain cereals were included in the Mediterranean diet score, while discouraging eggs and all dairy products regardless of fat content. Red meats, poultry, organ meats, and processed meats were discouraged but not excluded. In contrast, our HPDS score, aligned with the clinical recommendations, emphasizes the consumption of fish and seafood, eggs, low-fat dairy products, and minimally processed plant-based foods, including whole grains and legumes, while excluding red meats, poultry, and processed meat.

Our study found that individuals with the highest HPDS, indicating lower intake of saturated and partially hydrogenated fats, had a reduced risk of CVD events, aligning with previous research linking animal and ultra-processed food consumption, as well as saturated and trans-fat intake, to the development of IHD [49,50]. The highest HPDS, characterized by a rich dietary fiber and sterols content, lower energy density, and favorable nutrient profile, positively impacted serum cholesterol levels, glycemic control, body weight, and other cardiometabolic markers [51,52,53]. Increased fish consumption in the highest HPDS provided beneficial long-chain omega-3 fatty acids associated with a reduced risk of IHD [54,55]. The differences in protein sources between the lowest and highest HPDS further supported these findings, with plant-based proteins associated with a lower risk of obesity and T2DM compared to animal proteins rich in branched-chain amino acids [56]. Additionally, the higher intake of potassium, magnesium, folate, beta-carotene, vitamin C, E, and B12, coupled with lower sodium intake without compromising zinc intake, may further promote cardiometabolic health. This underscores the importance of overall diet quality in promoting cardiovascular health, beyond just focusing on fish consumption or excluding meats.

In our study, we did not find a significant effect of a healthy plant-based diet on stroke incidence or mortality, which is consistent with previous findings [5,6,7,52]. Tong et al. [6] reported a 43% higher risk of hemorrhagic stroke among vegetarians, even after adjusting for cardiometabolic risk factors. This association could potentially be attributed to the relatively low levels of LDL-cholesterol among the vegetarian participants in Tong’s [6] and other prospective studies [53,54]. It is hypothesized that low LDL-cholesterol concentrations may contribute to the necrosis of smooth muscle cells in the arterial wall’s medial layer, resulting in impaired endothelium and increased susceptibility to microaneurysms, ultimately increasing the risk of hemorrhage [54].

Our study has significant strengths, including the use of a large prospective cohort study with extensive data collection and long-term follow-up. We developed and validated a novel diet index, the Heart-Protective Diet Score, based on the clinical guidelines, which comprehensively evaluates diet quality. The use of detailed and accurate 24 h dietary data minimized misclassification of diet types, setting our study apart from other UK Biobank studies [5,6,7,57]. Our outcome assessment methods were robust, including medical record linkage and sensitivity analyses. Notably, the baseline cardiometabolic profiles were comparable across different HPDS quartiles, reducing potential confounding from cardiovascular risk factors. However, there are limitations to consider, such as the predominantly Caucasian study population, which may limit generalizability to other ethnic groups. Also, the use of a single 24 h recall inevitably results in the measurement error of long-term intake and intra-individual variations. However, to address this, we calculated the HPDS at baseline and the “averaged heart-protective diet score” based on repeated measurements and found a moderate correlation between them. We conducted further sensitivity analyses and confirmed the results generated from multiple 24 h recalls were almost identical to findings from the single 24 h recall. This finding is supported by another study [58] demonstrating good agreement for all food intakes in the UK Biobank cohort who completed 24 h recall more than once, with a substantial agreement for fish, meat, vegetable, and fruit intakes after 4 years. Despite the results, this study has several limitations: (i) As an observational study, we cannot establish causality, and there is a possibility of residual confounding from both dietary and non-dietary factors. Nevertheless, we have adjusted our analysis for several potential confounders related to lifestyle and health-conscious behaviors, which may have influenced the observed cardiovascular outcomes. (ii) Even though we used detailed and accurate 24 h dietary data, potential recall bias may still have influenced the results, and daily energy intake is not included in our analysis. (iii) The sample mainly comprised English individuals of white ethnicity, limiting the generalizability of our results to other populations.

Our HPDS demonstrates strong potential for application in both research and clinical contexts. Aligned with the most current cardiovascular and metabolic guidelines, the HPDS offers a practical and evidence-based tool for assessing diet quality among individuals with, or at elevated risk of, cardiovascular disease. In clinical practice, it could support personalized dietary counselling and facilitate the monitoring of dietary changes over time, particularly in primary care and preventative health settings. In research, the HPDS can be employed in epidemiological studies to evaluate adherence to heart-healthy dietary patterns and their associations with cardiovascular and metabolic outcomes. Additionally, it could be used to stratify participants according to baseline dietary quality and to measure changes in clinical nutrition trials in response to dietary interventions targeting improving cardiovascular health. The insights gained from such applications may help inform the development of targeted public health strategies for cardiovascular disease prevention.

## 5. Conclusions

In summary, findings from this large prospective cohort study revealed a robust link between adherence to a healthy diet score based on the 2021 AHA/ACC guidelines and a substantial decrease in cardiovascular mortality and morbidity, particularly for HF and IHD. These findings emphasize the significance of overall dietary quality in relation to cardiometabolic health, rather than focusing exclusively on specific diet types. Further research in diverse ethnic populations is necessary to validate our findings and enhance their generalizability. High-quality clinical trials delivering a healthy heart-protective plant-based diet in individuals at high risk are warranted to further strengthen the evidence for such a diet in the primary prevention of CVDs.

## Figures and Tables

**Figure 1 nutrients-17-02675-f001:**
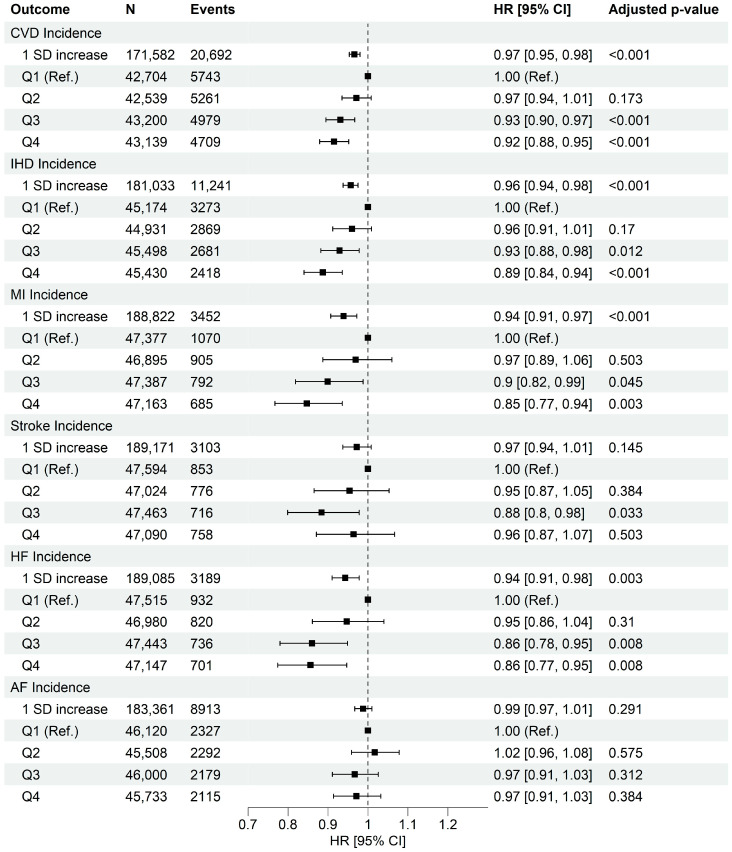
Association between quartiles of heart-protective diet score and risk of incident cardiovascular diseases (n = 192,274). Data were presented as adjusted hazard ratio [95% confidence interval]. Q1 was used as the reference group. Cox proportional hazard models were adjusted by age (timescale), sex, ethnicity, Townsend deprivation index, average house income, education level, dietary supplement use, history of cancer diagnosis, history of hyperglycemia/type 2 diabetes diagnosis, physical activity level, sitting, sleep quality, smoking habits, and alcohol intake. CVD, Cardiovascular Disease; IHD, Ischemic Heart Disease; MI, Myocardial Infarction; HF, Heart Failure; AF, Atrial Fibrillation; HR, Hazard Ratio; SD, Standard Deviation. Adjusted *p*-values were obtained using Benjamini–Hochberg procedure.

**Figure 2 nutrients-17-02675-f002:**
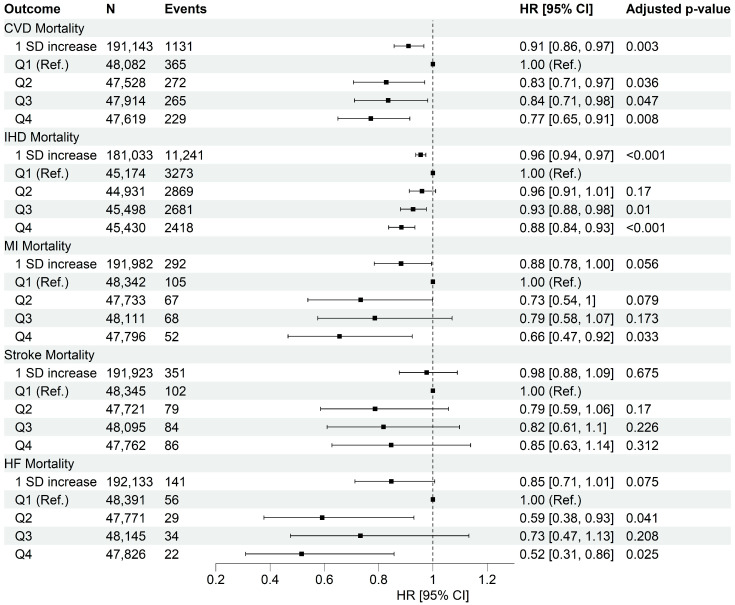
Association between quartiles of heart-protective diet score and risk of cardiovascular mortality. Data were presented as adjusted hazard ratio [95% confidence interval]. Q1 was used as the reference group. Cox proportional hazard models were adjusted by age (timescale), sex, ethnicity, Townsend deprivation index, average house income, education level, dietary supplement use, history of cancer diagnosis, history of hyperglycemia/type 2 diabetes diagnosis, physical activity level, sitting, sleep quality, smoking habits, and alcohol intake. CVD, Cardiovascular Disease; IHD, Ischemic Heart Disease; MI, Myocardial Infarction; HF, Heart Failure; HR, Hazard Ratio; SD, Standard Deviation. Adjusted *p*-values were obtained using Benjamini–Hochberg procedure.

**Figure 3 nutrients-17-02675-f003:**
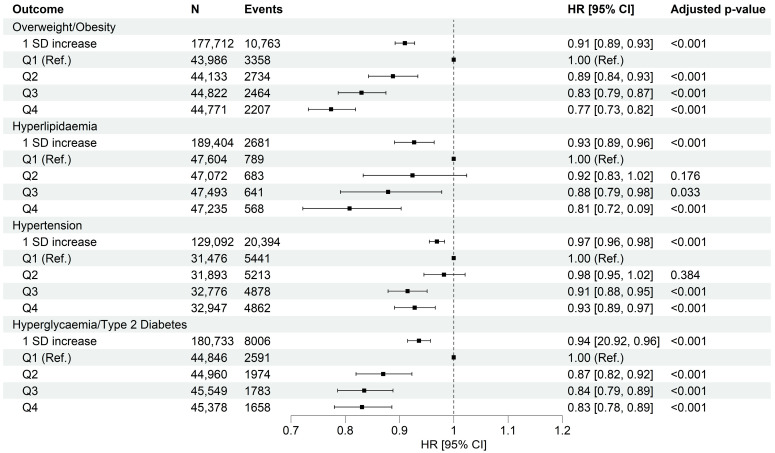
Association between quartiles of heart-protective diet score and risk of cardiometabolic abnormalities. Data presented as adjusted hazard ratio [95% confidence interval]. Q1 was used as the reference group. Cox proportional hazard models were adjusted by age (timescale), sex, ethnicity, Townsend deprivation index, average house income, education level, dietary supplement use, history of cancer diagnosis, history of hyperglycemia/type 2 diabetes diagnosis, physical activity level, sitting, sleep quality, smoking habits, and alcohol intake. HR, Hazard Ratio; SD, Standard Deviation. Adjusted *p*-values were obtained using Benjamini–Hochberg procedure.

**Table 1 nutrients-17-02675-t001:** Food items in the 22 food groups.

Food Groups	Food Items	Scoring
Wholegrains (14 items)	Porridge, muesli, oat crunch, bran cereal, non-white bread (flour types: brown, wholemeal, other type), seeded or other bread, crispbread, whole-wheat cereal, other cereal, oatcakes, wholemeal pasta, brown rice, couscous, other cooked grains (such as bulgur).	+
Fruits (20 items)	Avocado, mixed fruit, apple, banana, berries, cherries, grapefruit, grapes, mango, melon, orange, orange-like small fruits (such as satsuma), peach/nectarine, pear, pineapple, plum, other fruits, stewed/cooked fruit, prunes, other dried fruit	+
Non-starchy Vegetables (26 items)	Mixed vegetables, vegetable pieces, coleslaw, side salad, beetroot, broccoli, cabbage/kale, carrots, cauliflower, celery, courgette, cucumber, garlic, leeks, lettuce, mushrooms, onion, parsnip, sweet peppers, spinach, sprouts, fresh tomatoes, cooked or tinned tomatoes, turnip/swede, watercress, other vegetable intake	+
Starchy Vegetables (3 items)	Butternut squash, sweetcorn, sweet potato	+
Nuts and Seeds (5 items)	Unsalted peanuts, unsalted nuts, seeds	+
Legumes and Beans, Other Vegetarian Protein Alternatives (9 items)	Beans, other beans or lentils (pulses), broad beans, green beans, peas, tofu, quorn	+
Uncoated Fish and Seafood (7 items)	Tinned tuna, oily fish, white fish, prawns, lobster/crab, shellfish, other fish intake	+
Eggs (5 items)	Whole eggs, omelettes or scrambled eggs, eggs in sandwiches, scotch egg, other egg dishes	+
(Reduced-fat and/or No Added Sugar) Milk and Dairy Products (4 items)	Milk, low fat hard cheese, low fat cheese spread, cottage cheese	+
Tea, Coffee and Other Low-calorie Drinks (11 items)	Instant coffee, filtered coffee, cappuccino, latte, espresso, other coffee drinks, standard tea, rooibos tea, green tea, herbal tea, other tea	+
Homemade soup (1 item)	Homemade soup	+
Refined grains and cereals, including discretionary choices (12 items)	Sweetened cereal, plain cereal, white sliced bread, white bread, white bap, white bread roll, naan bread, garlic bread, white pasta, white rice, pancake, snackpot	−
Potatoes (4 items)	Fried potatoes, boiled/baked potatoes, mashed potatoes, crisps (e.g., potato chips)	−
Meat, Poultry, and Processed Meat (10 items)	Sausage, beef, pork, lamb, crumbed or deep-fried poultry, poultry, bacon, ham, liver, other meat intake	−
Coated Fish and Seafood (2 items)	Breaded fish, battered fish	−
(Full-fat and/or Added Sugar) Milk and Dairy Products (10 items)	Flavored milk, yogurt, hard cheese, soft cheese, blue cheese, cheese spread, feta cheese, mozzarella cheese, goat’s cheese, other cheese	−
Processed Soup (2 items)	Powdered/instant soup intake, canned soup intake	−
Sugar, Sweets and Desserts, Cookies, and Pastries (23 items)	Intake of sugar added to coffee/tea/cereal, ice-cream, milk-based pudding, other milk-based pudding, soya dessert, fruitcake, cake, doughnuts, sponge pudding, cheesecake, other dessert, chocolate bar, white chocolate, milk chocolate, dark chocolate, chocolate-covered raisin, chocolate sweet, diet sweets, chocolate-covered biscuits, chocolate biscuits, sweet biscuits, cereal bar, other sweets	−
Savory Snacks (2 items)	Savory or cheesy biscuits, other savory snacks	−
Sugary Drinks including Juices and Sugar-sweetened Beverages (3 items)	Low-calorie hot chocolate, hot chocolate, and other non-alcoholic drinks	−
Artificial sweetener (1 item)	Artificial sweetener added to coffee/tea/cereal	−
Unhealthy Fat (with/without Carbohydrates) (2 items)	Butter/margarine on bread/crackers (spreadable, low fat, normal fat, or unknown type), number of bread slices/baps/bread rolls/crackers/crispbreads/oatcakes/other bread types with butter/margarine	−

Food groups denoted by a +/− scoring are positively/negatively correlated with the healthy predominantly plant-based heart-protective diet score.

**Table 2 nutrients-17-02675-t002:** Baseline characteristics of participants based on quartile categories of heart-protective diet score.

	Heart-Protective Diet Score Quartiles			
	Q1 (n = 48,453)	Q2 (n = 47,778)	Q3 (n = 48,012)	Q4 (n = 48,031)
Heart-protective diet score, mean (SD)	−6.8 (3.6)	0.3 (1.5)	5.2 (1.5)	12.5 (3.8)
Age at recruitment, years, mean (SD)	55.3 (8.2)	56.1 (8.0)	56.7 (7.8)	57.0 (7.6)
Sex, male, N [%]	26,619 [54.9]	21,435 [44.9]	18,965 [39.5]	16,283 [33.9]
Ethnicity, White, N [%]	46,212 [95.4]	45,654 [95.6]	45,977 [95.8]	45,877 [95.5]
Smoking status, N [%]				
Never	26,818 [55.3]	27,386 [57.3]	28,022 [58.4]	28,192 [58.7]
Former	16,313 [33.7]	16,460 [34.5]	16,710 [34.8]	17,000 [35.4]
Current	5195 [10.7]	3806 [8.0]	3176 [6.6]	2724 [5.7]
Missing	127 [0.3]	126 [0.3]	104 [0.2]	115 [0.2]
Townsend deprivation index, mean (SD)	−1.5 (3.0)	−1.7 (2.9)	−1.7 (2.8)	−1.6 (2.9)
College or university degree, Yes, N [%]	16,714 [34.5]	20,026 [41.9]	22,205 [46.2]	24,424 [50.9]
Average total household income before tax, N [%]				
<£18,000	7119 [14.7]	6302 [13.2]	5923 [12.3]	6017 [12.5]
£18,000–30,999	10,740 [22.2]	10,239 [21.4]	10,220 [21.3]	10,192 [21.2]
£31,000–51,999	12,937 [26.7]	12,195 [25.5]	12,374 [25.8]	12,173 [25.3]
£52,000–100,000	10,168 [21]	10,913 [22.8]	11,097 [23.1]	11,091 [23.1]
>£100,000	2521 [5.2]	3270 [6.8]	3397 [7.1]	3624 [7.5]
Physical activity score, N [%]				
Low	5965 [12.3]	5437 [11.4]	4660 [9.7]	3910 [8.1]
Moderate	19,019 [39.3]	19,082 [39.9]	19,205 [40.0]	18,453 [38.4]
High	14,387 [29.7]	14,756 [30.9]	15,917 [33.2]	18,126 [37.7]
Sitting score (hours/day), mean (SD)	5.2 (2.5)	4.8 (2.3)	4.6 (2.2)	4.3 (2.2)
Sleep score [range: 0–5], mean (SD)	3.6 (1.0)	3.7 (1.0)	3.7 (1.0)	3.8 (1.0)
Dietary supplement use, yes, N [%]	22,261 [45.9]	24,244 [50.7]	25,842 [53.8]	27,855 [58.0]
Alcohol intake from past 24 h, g, mean (SD)	18.1 (26.4)	17.9 (25.0)	16.4 (23.1)	14.2 (21.2)
BMI, kg/m^2^, mean (SD)	27.5 (4.8)	27.0 (4.6)	26.6 (4.4)	26.2 (4.5)
SBP, mmHg, mean (SD)	137.1 (17.9)	136.6 (18.3)	136.5 (18.4)	136.1 (18.6)
DBP, mmHg, mean (SD)	82.6 (10.0)	82.0 (10.0)	81.6 (10.0)	81.2 (10.0)
TC, mmol/L, mean (SD)	5.7 (1.1)	5.8 (1.1)	5.8 (1.1)	5.8 (1.1)
LDL-C, mmol/L, mean (SD)	3.6 (0.8)	3.6 (0.8)	3.6 (0.8)	3.6 (0.8)
HDL-C, mmol/L, mean (SD)	1.4 (0.4)	1.5 (0.4)	1.5 (0.4)	1.6 (0.4)
Triglycerides, mmol/L, mean (SD)	1.8 (1.1)	1.7 (1.0)	1.6 (1.0)	1.6 (0.9)
HbA1c, mmol/mol, mean (SD)	35.6 (6.1)	35.4 (5.9)	35.3 (5.8)	35.3 (5.5)

Data are expressed as mean (SD) for continuous variables and N [%] for categorical variables, where % refers to the proportion in the column group. BMI, Body Mass Index; DBP, Diastolic Blood Pressure; HbA1c, glycated haemoglobin; HDL-C, high-density lipoprotein cholesterol; LDL-C, low-density lipoprotein cholesterol; SBP, Systolic Blood Pressure; TC, total cholesterol.

**Table 3 nutrients-17-02675-t003:** Food groups and nutrient intake based on quartile categories of heart-protective diet scores.

	Heart-Protective Diet Score Quartiles			
	Q1 (n = 48,453)	Q2 (n = 47,778)	Q3 (n = 48,012)	Q4 (n = 48,031)
Heart-protective diet score	−6.8 (3.6)	0.3 (1.5)	5.2 (1.5)	12.5 (3.8)
**Food groups intake (grams/week)**				
Wholegrains (1 serve: 60 g)	76.2 (98.8)	116.2 (107.5)	142.9 (110)	176.6 (117.1)
Fruits (1 serve: 80 g)	70.6 (100.9)	120.9 (127.7)	174.4 (145.5)	262.7 (173.6)
Non-starchy Vegetables (1 serve: 80 g)	102.7 (132.8)	156.6 (162.9)	211.1 (185)	311.3 (228.7)
Starchy Vegetables (1 serve: 75 g)	1.2 (8.7)	2.5 (13)	4.4 (17.2)	10.7 (27.7)
Nuts and Seeds (1 serve: 30 g)	1.9 (8.2)	3.2 (11.2)	4.8 (13.6)	9.3 (19.5)
Legumes and Beans, Other Vegetarian Protein Alternatives (1 serve: 80 g)	15 (34.7)	23.7 (43.8)	32.8 (50.6)	54.7 (66.1)
Uncoated Fish and Seafood (1 serve: 140 g)	14 (46.6)	27.8 (63.2)	43 (76.6)	72.7 (95.5)
Eggs (1 serve: 120 g)	23.6 (59.9)	29.8 (67.4)	34.5 (72.1)	47.7 (84.7)
(Reduced-fat and/or No Added Sugar) Milk and Dairy Products (1 serve: 250 mL)	17.2 (76.3)	23.5 (85.2)	32.5 (100.0)	52.8 (127.8)
Tea, Coffee and Other Low-calorie Drinks (1 serve: 250 mL)	345.2 (476.6)	506.2 (517.4)	611.1 (525)	781.8 (548.8)
Homemade soup (1 serve: 250 mL)	8.6 (48.1)	16.8 (66.8)	25.9 (83.6)	46.5 (112.1)
Refined grains and cereals, including discretionary choices (1 serve: 60 g)	114.6 (101.5)	60.7 (76.1)	39.5 (60.1)	23.5 (43.7)
Potatoes (1 serve: 75 g)	66.4 (64.5)	55.3 (56.5)	49 (52.0)	39 (45.9)
Meat, Poultry and Processed Meat (1 serve: 70 g)	109.6 (91.7)	85.9 (79.7)	70.4 (71.4)	49.1 (62.9)
Coated Fish and Seafood (1 serve: 140 g)	5.6 (24.1)	3.9 (18.9)	2.7 (15.7)	1.4 (11.0)
(Full-fat and/or Added Sugar) Milk and DairyProducts (1 serve: 250 mL)	329.4 (302.7)	296.2 (288.1)	285.1 (282)	268.8 (267.2)
Processed Soup (1 serve: 250 mL)	27.1 (82.4)	20.3 (70.9)	16.4 (63.9)	11.3 (53.1)
Sugar, Sweets and Desserts, Cookies and Pastries (1 serve: 40 g)	80.4 (79.5)	52.4 (63.6)	38.5 (54.1)	24.5 (42)
Savory Snacks (1 serve: 30 g)	2.2 (9.1)	1.4 (6.9)	1.1 (5.9)	0.8 (4.9)
Sugary Drinks including Juices and Sugar-sweetened Beverages (1 serve: 150 mL)	21.1 (70.1)	14.6 (56.3)	12.3 (50.7)	9.0 (44.5)
Artificial sweeteners (1 serve: 4 g)	1.4 (3.7)	0.9 (2.9)	0.6 (2.4)	0.4 (1.9)
Unhealthy Fat (with/without Carbohydrates) (1 serve: 10 g)	6.3 (11)	4.2 (9.1)	3.1 (7.9)	2.0 (6.7)
**Daily nutrient intake**				
Total energy intake, kcal/d	2187.3 (631.4)	2016.4 (604.0)	1970.7 (580.6)	1991.6 (578.6)
Total protein, g/d	81.6 (28.0)	78.4 (26.3)	78.4 (25.3)	81.9 (26.2)
Total fat, g/d	79.4 (31.0)	71.2 (29.8)	68.5 (29.1)	68.8 (30.2)
Saturated fatty acids, g/d	31.2 (13.8)	27.1 (12.6)	25.0 (12.0)	23.0 (11.6)
Monounsaturated fatty acids, g/d	28.7 (11.8)	25.6 (11.5)	24.7 (11.4)	25.3 (12.3)
*n*-3 fatty acids, g/d	1.8 (0.9)	1.8 (1.0)	1.9 (1.1)	2.3 (1.4)
*n*-6 fatty acids, g/d	10.8 (5.3)	10.3 (5.4)	10.5 (5.7)	11.7 (6.5)
Total carbohydrate, g/d	269.7 (85.4)	247.6 (81.9)	244.1 (80.0)	249.3 (80.6)
Total sugars, g/d	125.1 (54.1)	119.4 (50.9)	122.3 (50.2)	132.1 (52.2)
Dietary fiber, g/d	14.6 (5.9)	16.3 (6.3)	18.3 (6.5)	22.3 (7.7)
Alcohol, g/d	18.0 (26.2)	17.7 (24.9)	16.5 (23.1)	14.42 (21.4)
Vitamin C, mg/d	96.8 (71.1)	115.7 (77.9)	134.8 (83.4)	169.8 (97.5)
Vitamin E, mg/d	10.5 (5.0)	10.17 (4.9)	10.65 (4.9)	12.3 (5.2)
Vitamin B12, μg/d	5.7 (3.7)	5.80 (3.7)	6.1 (3.8)	6.9 (4.2)
Folate, μg/d	277.0 (107.3)	292.7 (110.0)	315.5 (110.4)	364.1 (122.7)
Beta carotene, μg/d	1727.4 (2089.6)	2239.4 (2445.7)	2760.2 (2724.9)	3937.7 (3566.4)
Iron, mg/d	11.7 (4.0)	11.8 (4.0)	12.2 (4.0)	13.5 (4.3)
Zinc, mg/d	9.8 (4.0)	9.5 (3.7)	9.5 (3.5)	9.8 (3.5)
Magnesium, mg/d	299.9 (90.3)	313.3 (92.5)	333.8 (93.5)	379.5 (106.4)
Iodine, μg/d	203.8 (109.3)	201.9 (111.5)	207.9 (119.6)	224.2 (135.7)
Sodium, mg/d	2158.5 (912.8)	1895.3 (845.3)	1814.8 (810.0)	1843.1 (820.2)
Potassium, mg/d	3342.3 (1099.5)	3478.5 (1100.8)	3686.0 (1088.8)	4128.9 (1157.0)

Data are expressed as mean (SD).

## Data Availability

We have our data ready to be accessed upon reasonable requests, and please note that the UK Biobank study requires investigators to register in the project to access the data, which are not publicly available. The cardiovascular outcomes and dietary UK Biobank data are available on application to the UK Biobank. This research has been conducted using the UK Biobank Resource under Application 62594. R scripts for analyses can be accessed via the online repository (https://github.com/sylviawt/HPDS_UKBB. DOI: 10.5281/zenodo.8115595). The manuscript’s guarantors (T.W. and A.N.) affirm that the manuscript is an honest, accurate, and transparent account of the study being reported; that no important aspects of the study have been omitted; and that any discrepancies from the study as planned (and, if relevant, registered) have been explained. This article is a revised and expanded version of a paper entitled [Heart-Protective Diet Scores, Cardiometabolic Risk and Cardiovascular Disease Incidence and Mortality: A Prospective Study From UK Biobank] [63], which was presented at [the American Society for Nutrition (ASN) conference 2023 in Boston].

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
