# Peer review of "Association of Healthy Predominantly Plant-Based Diet with Reduced Cardiovascular Disease Incidence and Mortality and Development of Novel Heart-Protective Diet Index†"

_nutrients, 2025, doi:10.3390/nu17162675_

Round 1

Reviewer 1 Report (Previous Reviewer 3)

Comments and Suggestions for Authors

Interesting idea of this study, my recommendations are the following:
Results- I recommend that the information mentioned in the tables should not be duplicated when interpreting, I recommend reviewing all subsections.
At the end of the Discussion section I recommend mentioning the practical implications of the study.

Author Response

Interesting idea of this study, my recommendations are the following:

  1. Results- I recommend that the information mentioned in the tables should not be duplicated when interpreting, I recommend reviewing all subsections.

Response: Thank you. We have carefully reviewed the Results and revised the text to minimize redundancy with the tables. We now focus on highlighting key findings and patterns. All relevant updates are shown in track changes in the manuscript.

  1. At the end of the Discussion section I recommend mentioning the practical implications of the study.

Response: Thank you. We have now expanded the Discussion section to clearly articulate the practical and clinical implications of the Heart-Protective Diet Score.

Practical Implications

Our HPDS demonstrates strong potential for application in both research and clinical contexts. Aligned with the most current cardiovascular and metabolic guide-lines, the HPDS offers a practical and evidence-based tool for assessing diet quality among individuals with, or at elevated risk of, cardiovascular disease. In clinical practice, it could support personalised dietary counselling and facilitate the monitoring of dietary changes over time, particularly in primary care and preventative health settings. In re-search, the HPDS can be employed in epidemiological studies to evaluate adherence to heart-healthy dietary patterns and their associations with cardiovascular and metabolic outcomes. Additionally, it could be used to stratify participants according to baseline dietary quality and to measure changes in nutrition clinical trials in response to dietary interventions targeting improving cardiovascular health. The insights gained from such applications may help inform the development of targeted public health strategies for cardiovascular disease prevention.

Reviewer 2 Report (Previous Reviewer 2)

Comments and Suggestions for Authors

The modifications made by the authors are adequate. I recommend this manuscript be published in Nutrients.

Author Response

#Reviewer 2

The modifications made by the authors are adequate. I recommend this manuscript be published in Nutrients.

Response: Thank you very much for your positive feedback. We are grateful for your thoughtful review and are pleased to hear that you find the revised manuscript suitable for publication.

Reviewer 3 Report (New Reviewer)

Comments and Suggestions for Authors

The article deals with the development and validation of a new diet quality index (Heart-Protective Diet Score – HPDS) and its association with cardiovascular disease (CVD) risk and mortality in the UK population. The aim of the study is justified and based on the need to develop a qualitative assessment of plant-based diets that include healthy animal products, which is a current and important public health issue.
An innovative approach to creating a nutrition index in line with current AHA, ESC and AHF guidelines.
Justified need – gaps in existing indices that only consider plant-based products.
Clearly defined objective and well formulated hypothesis.
There is no sound rationale for not including certain products, e.g. vegetable fats or alcohol – the authors refer to ‘variability in composition’ but not specific data.
The study was conducted on a large UK biobank sample (n=192,274) using data from the Oxford WebQ questionnaire (24-hour recall).
Very large study sample, which increases statistical power.
Long follow-up period (12.3 years on average).
High-quality statistical analysis: Cox model, FDR, sensitivity analysis, interaction tests, imputation of missing data.
The main dietary data come from a single 24-hour recall, which increases the risk of systematic errors.
The correlation coefficient between the baseline value and the average HPDS was only 0.53, which may indicate instability of the index.
Some variables may still act as confounders despite the control, e.g. a tendency towards healthy behaviour (healthy respondent bias).
The results indicate a statistically significant reduction in CVD risk and mortality in individuals with the highest HPDS.
Consistency of results across analyses: primary, secondary and sensitivity.
Comprehensive data – not only CVD but also metabolic risk factors.
Subgroup analysis (gender, deprivation index) provides additional context.
The magnitude of effects (e.g. HR=0.92 for CVD overall) can be considered moderate rather than groundbreaking.
The lack of an effect for stroke is not only consistent with previous studies, but should be discussed in more detail.
The authors effectively compare their index with existing dietary measures (PDI, hPDI, uPDI) and point out the advantages of the HPDS.
Reliable interpretation of the results in the light of previous studies.
Consideration of possible biological mechanisms.
Honest discussion of limitations.
Lack of reflection on the potential implications for clinical practise or health policy.
No example of an HPDS-compliant menu is provided, which limits its practical application.
Overall assessment:
The article makes an important contribution to research on diet quality and its impact on cardiovascular health and provides an innovative indicator that can be used in future epidemiological studies.
Recommendations:
 It is recommended:
expand the discussion on the practical application of the HPDSand provide an example of clinical application.

Author Response

  1. Overall assessment:
    The article makes an important contribution to research on diet quality and its impact on cardiovascular health and provides an innovative indicator that can be used in future epidemiological studies.

Response: We sincerely thank the reviewer for acknowledging the significance and innovation of our work. We appreciate your encouraging feedback.

  1. Recommendations:
    Expand the discussion on the practical application of the HPDS and provide an example of clinical application.

Response: Thank you. We have now expanded the Discussion section to clearly articulate the practical and clinical implications of the Heart-Protective Diet Score.

Practical Implications

Our HPDS demonstrates strong potential for application in both research and clinical contexts. Aligned with the most current cardiovascular and metabolic guide-lines, the HPDS offers a practical and evidence-based tool for assessing diet quality among individuals with, or at elevated risk of, cardiovascular disease. In clinical practice, it could support personalised dietary counselling and facilitate the monitoring of dietary changes over time, particularly in primary care and preventative health settings. In re-search, the HPDS can be employed in epidemiological studies to evaluate adherence to heart-healthy dietary patterns and their associations with cardiovascular and metabolic outcomes. Additionally, it could be used to stratify participants according to baseline dietary quality and to measure changes in nutrition clinical trials in response to dietary interventions targeting improving cardiovascular health. The insights gained from such applications may help inform the development of targeted public health strategies for cardiovascular disease prevention.

  1. There is no sound rationale for not including certain products, e.g. vegetable fats or alcohol – the authors refer to ‘variability in composition’ but not specific data.

Response: We thank the reviewer for this comment. We realized that we had omitted from the manuscript a clarification that, although alcohol intake was not included in the HPDS scoring algorithm, it was included as a covariate in our statistical models. We have now explicitly stated this in the text. In addition, we have revised the manuscript to add additional 8 references [Ref #26-33] to support our decision to exclude these items from the score. The manuscript has been updated accordingly.

  1. Figures and tables can be improved:

Response: Thank you. We have improved the figures and tables and we will work closely with the editorial team if it needs to be improved further.

This manuscript is a resubmission of an earlier submission. The following is a list of the peer review reports and author responses from that submission.

Round 1

Reviewer 1 Report

Comments and Suggestions for Authors

Dear authors,

 I found the topic interesting.

However, the article must be revised carefully.

1. Please format all the references and tables according to MDPI guidelines.

2. Check carefully the first sentence on page 5 - these is a technical mistake. I underlined it in yellow. " The primary outcomes included the CVD incidence and mortality. We defined CVD

outcomes based on the earliest diagnosis recorded according to the International Classifi-

cation of Disease 10th revision (ICD10) (see Table S10) for the following conditions: i)..."

3. The data is confusing. What do you mean with "plant-based" diet? Vegetarian diet or vegan diet or both ? This must be explained.

4. Table 1. Food items in the 22 food groups.

Could you include an explanation how did you decide to create these groups and why?

5. Discussion: You could support the benefits of adopting plant-based diets from other studies. Please provide data from studies investigating the benefits from vegan/vegetarian diet. There are many studies. Moreover, could you compare your study to similar studies?

Do you have any data about the daily calorie intake of the participants in the study?

6. Could you include a new section "Limitations of the study"? It would be nice if you explain the limitations of the study.

Author Response

Please see attached the word document for the author's reply.

Reviewer 2 Report

Comments and Suggestions for Authors

Before this manuscript can be considered for publication in Nutrients I suggest the following modifications:

The abstract should be structured. Conclusions should be improved and the authors are encouraged to include some directions for further actions/investigations.

References should be formatted according to the journal’s guidelines.

Line numbering is missing.

The “LAY SUMMARIES” section should be removed.

Sections and subsections should be numbered. The authors should pay attention to the journal’s instructions.

The study’s aims should be better clarified at the end of the introductory section.

In the Methods section, the authors should better explain their options for the selected tools used in the research.

Tables should be adequately formatted.

A stronger Discussion should be provided. More citations are encouraged to compare the obtained results with others. The study’s limitations should be better discussed as well.

The Conclusions are adequate.

Author Response

(The authors gave the same response as above.)

Reviewer 3 Report

Comments and Suggestions for Authors

Interesting idea of ​​this study, my recommendations are the following:
Abstract I recommend mentioning the average age and standard deviation of the subjects included in the study.
Introduction I recommend expanding by mentioning, to the target population, the typologies of diets on scientific bases.
The Methods section I recommend introducing a new section called Study design where to mention the typology of the study and other specific aspects.
Table 2 I recommend mentioning in the first column what the mentioned values ​​represent (n, % etc.) where applicable.
I recommend that the mention of bibliographic indexes be done in accordance with the editing rules.
I recommend that at the end of the Discussion section I mention the limits of this study.
I recommend expanding future research directions from the conclusions.

Comments on the Quality of English Language

-

Author Response

(The authors gave the same response as above.)

Round 2

Reviewer 1 Report

Comments and Suggestions for Authors

Dear authors,

According to your reply I do not consider that you have addressed all comments. I believe the quality of the manuscript must be improved.